# Comparative efficacy of Chinese herbal injections for treating severe pneumonia: A protocol for systematic review and Bayesian network meta-analysis of randomized controlled trials

**Lu Xiao**[1,2,3☯], **Liqing Niu**[1,2☯], **Xuemin Zhang**[1,2], **Xuezheng Liu**[1,2,3], **Xinqiao Liu**[1,2,3]*, **Chongxiang Sun**[1,2], **Xiaokun Yang**[1,2]

**1** Emergency Department, First Teaching Hospital of Tianjin University of Traditional Chinese Medicine, Tianjin City, Tianjin Province, China, **2** National Clinical Research Center for Chinese Medicine Acupuncture and Moxibustion, Tianjin City, Tianjin Province, China, **3** State Key Laboratory of Multi-fractions Traditional Chinese Medicine, Tianjin University of Traditional Chinese Medicine, Tianjin City, Tianjin Province, China

☯ These authors contributed equally to this work.
* 13821150112@163.com

**Data Availability Statement:** All relevant data are within the paper and its Supporting information files.

## Abstract

### Background

Severe pneumonia (SP) has a high mortality and is responsible for significant healthcare cost. Chinese herbal injections (CHIs) have been widely used in China as a novel and promising treatment option for SP. Therefore, this study will assess and rank the effectiveness of CHIs to provide more sights for the selection of SP treatment.

### Method

Seven databases will be searched, including PubMed, the Cochrane Library, Embase, Web of Science, China National Knowledge Infrastructure (CNKI), Wanfang Database, and the Chinese Scientific Journal Database (VIP) from their inception up to October, 2021. The literatures screening, data extraction and the quality assessment of included studies will be conducted independently by two reviewers. Then Bayesian network meta-analysis (NMA) will be performed by WinBUGS 14.0 and STATA 14.0 software. Surface under the cumulative ranking curve (SUCRA) probability values will be applied to rank the examined treatments. The risk of bias of each included study will be evaluated using the Revised Cochrane risk-of-bias tool for randomized trials (ROB 2). Publication bias will be reflected by a funnel plot.

### Results

The results of this NMA will be disseminated through a peer-reviewed journal publication.

**Funding:** This work is supported by the support program of National Program on Key Basic Research Project (973 Program, No.2009CB522702) and First Teaching Hospital of Tianjin University of Traditional Chinese Medicine (No.201928). The funders had and will not have a role in study design, data collection and analysis, decision to publish, or preparation of the manuscript.

**Competing interests:** The authors have declared that no competing interests exist.

## Conclusion

Our study findings maybe reveal which CHI or CHIs will be better in the treatment of SP and provide more therapy strategies for clinical practitioners and patients.

## PROSPERO registration number

CRD42021244587.

## Strengths and limitations of this study

Bayesian network meta-analysis (NMA) can integrate direct evidence with indirect evidence of severe pneumonia treated by Chinese herbal injections to generate a clinically useful ranking of these regimens.

This NMA will address Chinese herbal injections for SP and its findings may help to provide more sights for selection of SP treatment.

Evidence drawn from an NMA is limited and should be interpreted with caution.

We only included studies in Chinese and English languages, which may increase the publication bias.

## Introduction

Pneumonia is a persistent and pervasive burden of disease. In the US, pneumonia continues to be associated with significant morbidity and mortality, accounting for more than 1 million hospital admissions per year [1]. In China, the prevalence of pneumonia in 2 weeks was 1.1‰ according to China Health Statistics Yearbook in 2013. Though not fatal, pneumonia can be severe. A fifth of the patients hospitalized for pneumonia need to be admitted to intensive care units (ICU)and a third of those require mechanical ventilation [2,3]. Severe pneumonia (SP) remains a major cause of mortality and has a high mortality up to 30%-50% [1,4,5]. In addition, due to combined antibiotics, long mechanical ventilation, and hospitalization time, SP is responsible for significant healthcare costs [6].

Currently, therapies for SP mainly depend on antibiotics, mechanical ventilation, and corticosteroid [7]. Antibiotic therapy is the backbone of the management of SP. It must be started on an empiric basis and often a combined therapy, since the causative agent is not identified in a considerable proportion of patients and the delay in the administration of adequate antimicrobials is clearly associated with higher mortality in patients [8]. However, adequate initial antibiotics may elevate the risk of antibiotic resistance and the mortality [9,10]. The severity of pneumonia is determined by interacting processes of immune resistance and tissue resilience such as anti-inflammatory [11]. Corticosteroids could inhibit the expression and action of many cytokines involved in the inflammatory response associated with pneumonia [12]. However, the use of corticosteroids in clinical practice with SP remains controversial as the presence of adverse effects [13,14]. Therefore, there is an urgent need for combined and adjunctive therapeutic options to improve outcomes.

In recent years, Chinese herbal injections (CHIs) as adjuvant treatments for SP are widely applied in China [15–17]. Even in the treatment of COVID-19, CHIs display more superiority [18,19]. Currently, several CHIs including Xuebijing injection (XBJ), Tanreqing injection (TRQ), Reduning injection (RDN), Xiyanping injection (XYP), Shenfu injection (SF), Shenmai injection (SM), and so on have been widely used in SP because of their remarkable effects.

Their efficacy has been evidenced with systematic reviews [20–22]. CHIs combined with conventional Western medicine (WM) can greatly improve clinical symptoms and interrupt the vicious cycle of inflammation onset by blocking the uncontrolled release of endogenic inflammatory mediators like IL-6, IL-8, and TNF-α [16,18,23,24]. In addition, a meta-analysis showed that CHIs had less adverse drug reactions and adverse drug events (ADRs/ADEs), and the majority of the ADRs/ADEs had been resolved after drug withdrawal [20]. However, the head-to-head clinical trials comparing the efficacy and safety of the six CHIs are lacking up to now. It is difficult to identify the most effective one for patients with SP when the direct evidence is not enough. As a new method of evidence-based medical statistical methods, network meta-analysis (NMA) extends principles of conventional meta-analysis to the evaluation of multiple treatments in a single analysis by combining the direct and indirect evidence [25,26]. Another major value of NMA is that it can rank each CHI according to its effectiveness, which is important for clinicians to make the best treatment choices. Therefore, the purpose of this study is to assess the clinical efficacy and safety of different CHIs combined with WM and provide more evidence for rational selection of CHIs for SP using NMA.

## Methods

### Study registration

This protocol has been prepared under the guidance of the Preferred Reporting Items for Systematic review and Meta-Analysis (PRISMA) Protocols guidelines [27], and it has been registered on PROSPERO platform (https://www.crd.york.ac.uk/prospero/) with an assigned registration number CRD42021244587.

### Ethics and dissemination

For all eligible studies were approved by local institutional review boards and ethical committees, and participants included were required to complete written informed consents, this study requires no further ethical approval.

### Eligibility criteria

The PICOS (participant, intervention, comparison, outcome and study design) principle has been applied in the study design.

**Type of included studies.** Randomized controlled trials (RCTs) regarding CHIs for the treatment of severe pneumonia will be included for analysis. There will be not limitations on language. Studies with non-RCT design, unobtainable data, duplicate publications and reviews will be excluded.

**Participants.** Adults (aged 18 years or older) with severe pneumonia, which should be confirmed according to the diagnostic criteria [28,29], patients with other critical diseases (tumor, pulmonary fibrosis, tuberculosis, and secondary respiratory failure of other systems) will be excluded. No limitations exist in gender, race, or nationality.

**Intervention.** All experimental groups must have been treated with CHI or CHI combined with conventional Western medicine. The control groups must have been treated with different types of CHI, or different types of CHI combined with conventional Western medicine, or only conventional Western medicine.

**Outcomes.** The primary outcome includes 28-day mortality and clinical effective rate. The 28-day mortality is death from any cause and is assessed 28 days after the start of treatment. The mortality at 28 to 30 days is considered equivalent to 28-day mortality. The clinical effective rate is calculated by the following formula: (number of cured patients + number of

improved patients)/total number of patients × 100%. Patients are regarded as cured when their clinical symptoms disappear and the objective indicators return to normal. Patients are regarded as improved when their clinical symptoms alleviate and the objective indicators are improved. If the clinical symptoms and objective indicators are either unchanged or aggravated, the patients are considered to be an invalid effectiveness status. The secondary outcomes include length of stay in ICU, duration of mechanical ventilation, C-reactive protein (CRP), procalcitonin (PCT), leukocyte (WBC) and ADRs/ADEs.

## Data sources

The study search will be mainly based on electronic databases, including PubMed, the Cochrane Library, Embase, Web of Science, China National Knowledge Infrastructure (CNKI), Wanfang Database, and the Chinese Scientific Journal Database (VIP) from their inception up to October, 2021. The medical subject headings (MeSH) and free text words will be used. Language restriction do not exist in this study. Search terms are severe pneumonia, Chinese herbal injection and others. Full details of the search strategy in PubMed are shown in Table 1 and other strategies for the remaining databases are shown in supporting information (S2).

## Study selection and data extraction

Two researchers (LQ Niu and L Xiao) will independently screen the studies according to thee-ligibility criteria. After checking for duplicate studies, the researchers will eliminate reviews and irrelevant studies by reading the titles and abstracts. Then, the full text will be read to select studies that meet the eligibility criteria. The reasons for exclusion of trials in full-text review will be reported in the excluded studies list. Any disagreement will be resolved by discussion or a third researcher (XZ Liu). The references of included studies will be tracked to identify other relevant studies. A data spreadsheet will be developed with Microsoft Excel 2019 to

**Table 1. PubMed search strategy.**

| Number | Search terms |
|---|---|
| #1 | Severe pneumonia[MeSH Terms] |
| #2 | Severe pneumonia[Title/Abstract] |
| #3 | #1 OR #2 |
| #4 | Chinese herbal injection[MeSH Terms] |
| #5 | Chinese herbal injection[Title/Abstract] |
| #6 | Traditional Chinese medicine injection[Title/Abstract] |
| #7 | Traditional Chinese medicine[Title/Abstract] |
| #8 | xuebijing[Title/Abstract] |
| #9 | tanreqing[Title/Abstract] |
| #10 | xiyanping[Title/Abstract] |
| #11 | reduning[Title/Abstract] |
| #12 | shenfu[Title/Abstract] |
| #13 | shenmai[Title/Abstract] |
| #14 | #4 OR #5 OR #6 OR #7 OR #8 OR #9 OR #10 OR #11 OR #12 OR #13 |
| #15 | randomized controlled trial[Publication Type] |
| #16 | controlled clinical trial[Publication Type] |
| #17 | random*[All Fields] |
| #18 | #15 OR #16 OR #17 |
| #19 | #3 AND #14 AND #18 |

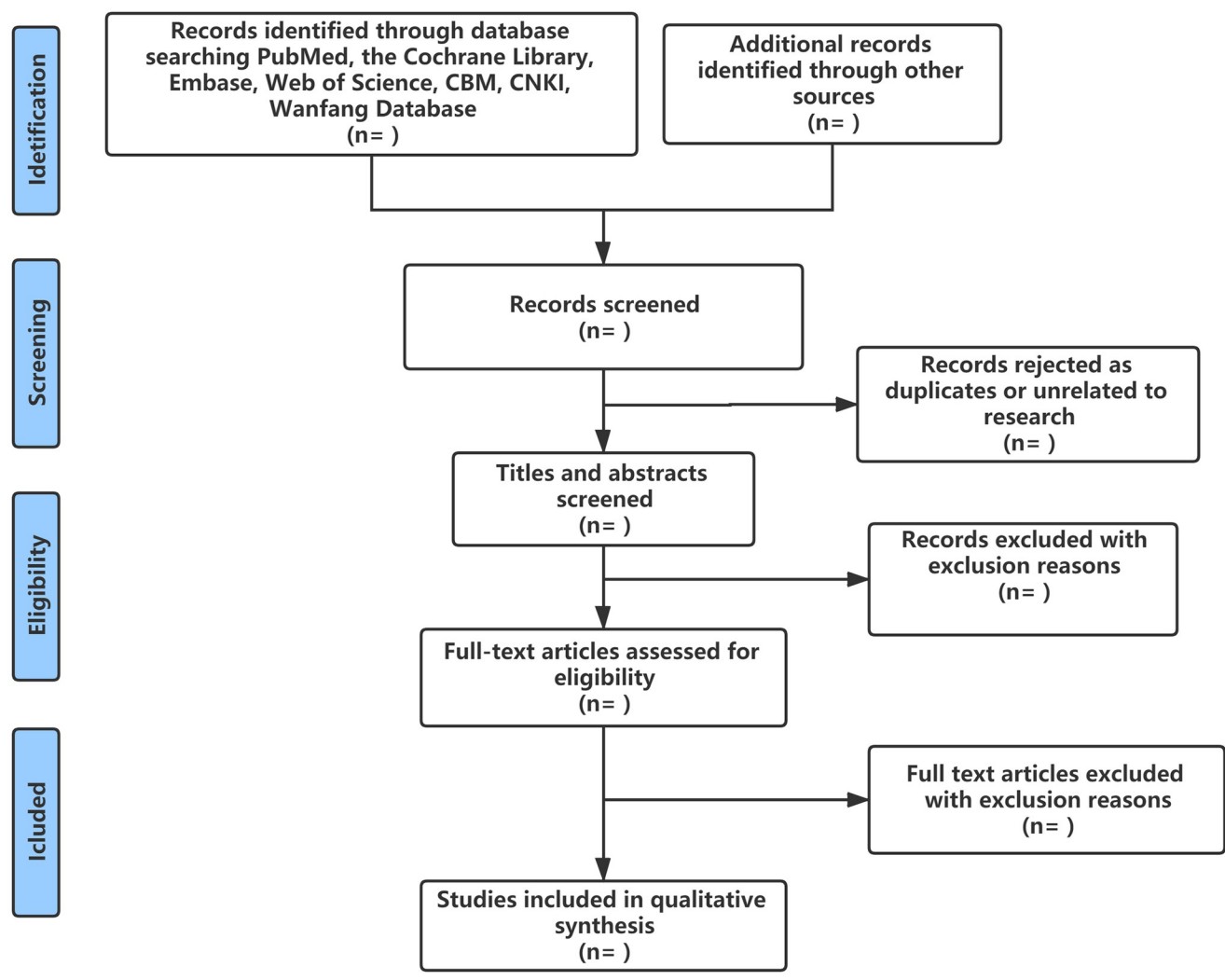

**Fig 1. Flow diagram of study inclusion.**

collect relevant information. The information, including eligible studies characteristics (e.g., first author, year of publication, country where the study was conducted), participant characteristics (e.g., gender, age, sample), interventions (e.g., name of CHI, duration, frequency of drugs) and outcomes will be extracted and entered into the spreadsheet. The literature selection process is illustrated in Fig 1.

## Risk of bias assessment

The risk of bias of each included study will be evaluated with Revised Cochrane risk-of-bias tool for randomized trials (ROB 2) [30]. The domains include the following: (1) randomisation process; (2) deviations from intended interventions; (3) missing outcome data; (4) measurement of the outcome; (5) selection of the reported result. There are some signalling questions required to answer "Yes (Y)", "Probably Yes (PY)", "Probably No (PN)", "No (N)", or "No Information (NI)" for each domain. After that, the risk of bias is categorized into 3 levels: high risk, some concerns, and low risk. These domain-level judgements will inform an overall risk

of bias judgment for the outcome. The risk of bias assessment will be performed by two independent reviewers (LQ Niu and L Xiao), and disagreements will be resolved by consensus or a third opinion.

## Assessment of the certainty of evidence

The overall certainty of evidence for each outcome collected will be assessed using the five GRADE considerations (bias of the trials, consistency of effect, imprecision, indirectness, and publication bias) by two reviewers. In addition, intransitivity and incoherence should be considered as well. Four steps will be utilized to assess the certainty of treatment effect estimates from NMA including direct and indirect treatment estimates for each comparison of the evidence network, rating the quality of each direct and indirect effect estimate, presenting the NMA estimate for each comparison of the evidence network and rating the quality of each NMA effect estimate [31]. The certainty of evidence will be rated 'high', 'moderate', 'low' and 'very low' according to the GRADE rating standards. Then, a minimally contextualised framework of GRADE approach will be used to draw conclusions from the NMA. The framework includes five steps involving choosing reference intervention and decision threshold, first classification of interventions based on comparison with reference, second classification based on comparisons between pairs of interventions, separating interventions into two main groups according to certainty of evidence, and checking consistency with pairwise comparisons and rankings [32]. Finally, we will communicate our findings using language that GRADE suggests to convey higher certainty (interventions are among the most effective) and language appropriate for lower certainty (interventions may be among the most effective) [33].

## Data synthesis

**Dealing with missing data.**   If required data is elusive or missing, the reviewer will contact the corresponding author of the original study by email. If data remains unobtainable, a systematic narrative review will be conducted through qualitative analysis. The potential impact of missing data will be assessed with a sensitivity analysis.

**Measures of treatment effect.**   For binary outcomes, the combined results will be calculated as odds ratios (ORs) and 95% credible intervals (95% CIs). For continuous outcomes, mean differences (MD) or standardized mean differences (SMD) and their 95% CIs will be used. When the 95% CIs of the ORs do not include one and the 95% CIs of the MDs or SMDs do not contain zero, the differences between the groups will be considered statistically significant.

**Assessment of heterogeneity.**   Clinical and methodological heterogeneity will be evaluated by closely checking the features of the participants, interventions, outcome measures, methods, and comparing fit of the fixed effect model and random effect model [34]. The Cochrane Q statistic and $I^2$ statistic will be employed to assess statistical heterogeneity between different studies [35]. $I^2$ value of 25%, 50%, and 75% is representative of low, moderate, and high heterogeneity, respectively. If the heterogeneity exists significantly ($P<0.05$, $I^2>50\%$), the sources of heterogeneity will be explored by subgroup analyses or meta regression. If we can't identify the cause of heterogeneity, a narrative analysis will be conducted [34].

**Assessment of similarity and consistency.**   An assessment of similarity and consistency will be performed to produce a credible and valid result. Since it is difficult to determine similarity using statistical analysis, the assessment will be based on clinical and methodological characteristics, including study designs, participant characteristics and interventions. If exist closed loops, we will use the loop-specific approach or the node-splitting approach to examine the consistency between direct and indirect evidences. If the inconsistency factor (IF) values

and their 95% CIs are truncated at zero in the loop-specific approach or *P*>0.05 in the node-splitting approach, they indicate that the 2 sources are in agreement [36,37]. Otherwise, it exists inconsistency between direct and indirect evidences. If so, we will revisit the studies to assess again the plausibility of the similarity assumption. If we identify possible reasons for this inconsistency, we will account for it by performing subgroup analyses or meta regression. If we can't identify the cause of inconsistency, we will refrain from performing an NMA [38].

**Network meta-analysis.** WinBUGS 14.0 (MRC Biostatistics Unit, Cambridge, UK) and STATA 14.0 (Stata Corporation, College Station, TX, USA) software will be employed to compute calculations and prepare graphs. The Markov chain Monte Carlo method will be performed by using the WinBUGS to carry out the NMA. In WinBUGS, the number of iterations will be set to 300 000, and the first 100 000 iterations will be used for the annealing algorithm to eliminate the impact of the initial value. The network graph will be constructed using STATA software to show a comparative relationship between different interventions. Surface under the cumulative ranking curve (SUCRA) probability values will be applied to rank the examined treatments, and the SUCRA value of 100% and 0% will be assigned to the best and worst treatments, respectively [39–41]. Furthermore, a comparison clustering analysis will be utilized to compare the effect of CHIs between different outcomes.

**Assessment of publication bias.** For each treatment comparison, the comparison-adjusted funnel plot will be used to assess the presence of small-study effects and publication bias if more than 10 studies are included. And its asymmetry will be evaluated with Egger's test [34,42].

**Sensitivity analysis.** We will perform sensitivity analysis to assess the robustness of the pooled results on outcomes according to sample size, missing data, or risk of bias (excluding studies with "high risk" as determined using the Revised Cochrane risk-of-bias tool for randomized trials) [30]. We will remove each study from the meta-analysis one at a time and recalculate the summary effect. A study will be considered influential if its removal changes the magnitude of the pooled effect by >10% [43].

## Discussion

Severe pneumonia, an important public health problem, is the leading infectious cause of death worldwide. For severe pneumonia, adequate initial antibiotics and long use in patients have increase the risk of antibiotic resistance. A number of studies have shown that the Chinese herbal injections have effects of decreasing the risk of antibiotic resistance by shortening the length of stay in ICU, duration of mechanical ventilation and so on [15–17,44–46]. Two systematic reviews have been conducted to assess the effectiveness of Chinese herbal injections for pneumonia [20,47]. However, as the worldwide spread of COVID-19, SP is given to more attention as its high mortality. Compared with community-acquired pneumonia (CAP), SP is more serious and requires to be treated in ICU. One of the studies only included elderly patients with pneumonia [47]. SP is associated with substantial morbidity and mortality, particularly in older adults or those with comorbid conditions. Nevertheless, SP can also occur in previously healthy young subjects originating distressing situations for its poor outcome [48]. Because of different disease severity and different ages of included participants, the CHIs being analyzed will be different compared with the other two reviews. Our study will include three extra injections: Xuebijing injection, Shenfu injection and Shenmai injection which are widely used in either CAP or SP and proved to be effective by lots of RCTs [15–17,49]. In addition, for SP, patients are mostly in the stage of immunosuppression. According to the theory of traditional Chinese medicine, those patients are diagnosed with deficiency syndrome. Therefore, Shenfu injection and Shenmai injection which were not included by the two previous reviews

will be involved for analysis in our study. What's more, the risk of bias will be evaluated with the ROB 2 and the overall quality of evidence for each outcome collected will be assessed using the GRADE. Hence, in order to generate reliable evidence based on a larger scale as compared with limited types of single studies, we will conduct a Bayesian NMA to combined direct and indirect comparisons of CHIs treating SP and rank the effectiveness and safety of the different CHIs. We expect to provide credible evidence and support of CHIs utility reasonably for clinical practitioners and patients in SP decision-making.

## Supporting information

**S1 Checklist. PRISMA-P 2015 checklist.**
(DOC)

**S1 Appendix. Search strategy.**
(DOCX)

**S1 Dataset. The draft of minimal dataset.**
(DOCX)

## Acknowledgments

### Ethics and dissemination

This review does not require ethics approval and the results of the NMA will be submitted to a peer-review journal.

## Author Contributions

**Conceptualization:** Lu Xiao, Liqing Niu.

**Data curation:** Xuemin Zhang, Chongxiang Sun.

**Formal analysis:** Liqing Niu, Xuemin Zhang.

**Funding acquisition:** Lu Xiao, Xinqiao Liu.

**Investigation:** Xuemin Zhang, Chongxiang Sun.

**Methodology:** Lu Xiao, Xuezheng Liu, Xiaokun Yang.

**Resources:** Lu Xiao, Xuemin Zhang, Xuezheng Liu.

**Software:** Liqing Niu, Chongxiang Sun.

**Supervision:** Xinqiao Liu, Xiaokun Yang.

**Writing – original draft:** Lu Xiao, Liqing Niu.

**Writing – review & editing:** Xuezheng Liu, Xinqiao Liu.

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
