## [Decision Letter · Decision Letter 0]

28 Sep 2021

PONE-D-21-21402

Comparative efficacy of Chinese herbal injections for treating severe pneumonia: A protocol for systematic review and Bayesian network meta-analysis of randomized controlled trials

PLOS ONE

Dear Dr. Liu,

Thank you for submitting your manuscript to PLOS ONE. After careful consideration, we feel that it has merit but does not fully meet PLOS ONE’s publication criteria as it currently stands. Therefore, we invite you to submit a revised version of the manuscript that addresses the points raised during the review process.

We look forward to receiving your revised manuscript.

Kind regards,

Ivan D. Florez, MD, MSc, PhD

Academic Editor

PLOS ONE

2. Thank you for stating the following in the Acknowledgments/Funding Section of your manuscript:

“Funding acquisition: Lu Xiao, Xinqiao Liu”

“Funded studies:

This work is supported by the support program of National Program on Key Basic Research Project (973 Program, No.2009CB522702) and First Teaching Hospital of Tianjin University of Traditional Chinese Medicine (No.201928). Xinqiao Liu received  award of the former and Lu Xiao received the latter.

The funders had and will not have a role in study design, data collection and analysis, decision to publish, or preparation of the manuscript.”

3. Please include your tables as part of your main manuscript and remove the individual files. Please note that supplementary tables (should remain/ be uploaded) as separate ""supporting information"" files"""

4. We noticed you have some minor occurrence of overlapping text with the following previous publication(s), which needs to be addressed:

- https://journals.lww.com/co-pulmonarymedicine/Abstract/2017/05000/Pathogenesis_of_severe_pneumonia__advances_and.2.aspx

- https://www.tandfonline.com/doi/abs/10.1080/14787210.2018.1512403?journalCode=ierz20

- https://bmjopen.bmj.com/content/11/2/e039122.full

The text that needs to be addressed involves the introduction. 

In your revision ensure you cite all your sources (including your own works), and quote or rephrase any duplicated text outside the methods section. Further consideration is dependent on these concerns being addressed

Additional Editor Comments (if provided):

Your manuscript has been reviewed by two experts in the field, and they have found some points that need to be addressed before this manuscript is considered for publication. Please go through the reviewers' comments and consider addressing these points, and prepare a revised version. Although all the comments are important and need to be considered in the response, please pay particular attention to the firs comment from reviewer 2, who highlights that this work lacks of novelty.

Reviewers' comments:

Reviewer's Responses to Questions

**Comments to the Author**

1. Does the manuscript provide a valid rationale for the proposed study, with clearly identified and justified research questions?

Reviewer #1: Partly

Reviewer #2: Partly

2. Is the protocol technically sound and planned in a manner that will lead to a meaningful outcome and allow testing the stated hypotheses?

Reviewer #1: Partly

Reviewer #2: Partly

3. Is the methodology feasible and described in sufficient detail to allow the work to be replicable?

Reviewer #1: No

Reviewer #2: No

4. Have the authors described where all data underlying the findings will be made available when the study is complete?

Reviewer #1: Yes

Reviewer #2: No

5. Is the manuscript presented in an intelligible fashion and written in standard English?

Reviewer #1: No

Reviewer #2: No

6. Review Comments to the Author

You may also provide optional suggestions and comments to authors that they might find helpful in planning their study.

Reviewer #1: 1. The author restricted the search date up until April 2021, which’s been more than half year until now. Consider update this.

2. The authors listed six CHIs and stated that “the head-to-head clinical trials comparing the efficacy of the six CHIs are lacking up to now.” If these six CHIs are the main ones of current use, it’s better to integrate these terms into the Search Strategy (i.e. specific CHIs as free text words)

3. Cochrane risk of bias tool is tailored for risk of bias assessment. Suggest use risk of bias throughout the manuscript instead of methodological quality, as these two could be different in some aspects.

4. In the eligibility criteria, the primary outcome “clinical effective rate” is unclear. Would be good to see the exact definition. By the way, if SP is a leading infectious cause of death as the author said, why not to collect the mortality? Also, why not to collect the adverse event? How did the authors decide the importance of the outcomes for patients?

5. Not sure how will the authors use GRADE. Under “Evaluation of study quality and risk of bias”, it would be good to see a detailed description of the overall process. Also, NMA has more considerations than for conventional MA. From what the authors referred in citation 31, there is nothing about GRADE for NMA (the first paper was published in 2014 https://www.bmj.com/content/349/bmj.g5630). So it’s really unclear how will this work.

6. Under Data synthesis, “If data remains unobtainable, the reviewer will exclude the study”. This may not appropriate since this is not in your eligible criteria and will of course lead to more reporting bias. You may consider include and report them descriptively and explain why they are not included in the data synthesis.

7. Under “Measures of treatment effect.”: “All outcomes will be presented with their 95% credible intervals (95% CIs) and as well.” Is there anything missing?

8. Under “Network meta-analysis” “After that, Publication bias will be reflected by a funnel plot” better to explain when is it appropriate to conduct a funnel plot, as the number of studies is not always enough for this analysis.

9. For the limitations, why include only RCT is a limitation when qualified RCTs are among the highest level of evidence. Include Chinese and English may increase the “risk of bias”, do you mean “publication bias”?

10. The funders had and will not have a role in study design… Do you mean the funder had a role in the study design? Or it’s a mistake?

11. In the background, “The largest challenge in the next few years will be to diminish this high and unacceptable mortality rate.” Suggest delete as it didn’t add more useful information.

12. In the background, the authors mentioned the effectiveness and benefits of CHIs, what about the potential AEs of CHIs? Suggest to explain.

13. In the discussion, “The application of Chinese herbal injections could be promoted because of the risk of antibiotic resistance” is unclear. Do you mean because of the risk of antibiotic resistance from antibiotics? Otherwise, it reads like CHI has the risk.

14. What does “experimentally based evidence” mean in the Discussion?

15. The authors stated “…this study will be the first Bayesian NMA of CHIs for the treatment of SP”, is there any existing frequentist NMA? Better to explain this point. If yes, may need to state why it’s necessary to do a new one.

16. Under the assessment of homogeneity, “I2≤ P≥0.1, 50%), If with homogeneity ( a fixed effect model will be adopted; If with obvious heterogeneity (P<0.01, I2 >50%), a random effect model will be applied.” is inappropriate. The choice between a fixed-effect and a random-effects meta-analysis should never be made on the basis of a statistical test for heterogeneity. Suggest refer to the Cochrane handbook. https://handbook-5-1.cochrane.org/chapter_9/9_5_4_incorporating_heterogeneity_into_random_effects_models.htm

17. Under “Assessment of similarity and consistency.”. What’s the method used to generate inconsistency factor? I didn’t find relevant information from citation 33. In addition, there is no consideration of the incoherence of each NMA and corresponding option to deal with the possible incoherence.

18. There are several language problems, including grammar issue and the accuracy of the wording. For examples, 1) the conclusion in the abstract, “Our study findings will reveal which CHIs is the best” where CHIs should be CHI as the author assume there is only one best (By the way, how could we know there will not be two equal effective CHIs?); 2) “…can integrate direct evidence with indirect evidence from currently applied Chinese herbal injections into severe pneumonia (SP) to generate a clinically useful ranking …”, should from be “of”? What means “injections into SP (a disease)”? 3) mortality has already covered the meaning of “rate” so mortality itself is appropriate instead of “mortality rate” 4) mixed tense in one sentence such as “A fifth of the patients hospitalized for pneumonia need to be admitted to intensive care units (ICU), and a third of those require mechanical ventilation, while 21% of the patients from the community needed admission to the ICU and 26% of them needed mechanical ventilation. 5) “…the delay in the administration of adequate antimicrobials is clearly associated with mortality in patients…” do you mean “with higher mortality…”? 6) “criterias” under “Study selection and data extraction”. There are many others. This manuscript would benefit from further language editing.

Reviewer #2: I would like to thank the authors for the opportunity to review their manuscript “Comparative efficacy of Chinese herbal injections for treating severe pneumonia: A

protocol for systematic review and Bayesian network meta-analysis of randomized

controlled trials” This Network meta-analysis sought out conduct a network meta-analysis approach in order to conclude the optimal mode of Chinese herbal injection for improving well-being among patients with severe pneumonia. This study is similar to those presented by previous studies, and therefore lack novelty. Authors should substantial revision with this article and recommend addressing the following concerns:

1. This study lacks innovation. This manuscript is highly similarly to other systematic review, including disease types, analysis methods and research objects. A problem not fully addressed in discussion is mainly the similarity and the consistency evidence compared to the previous publications. I recommend that you can cite two studies for your information(1.《Huang Xingyue,Duan Xiaojiao,Zhu Yingli et al. Comparative efficacy of Chinese herbal injections for the treatment of community-acquired pneumonia: A Bayesian network meta-analysis of randomized controlled trials.[J] .Phytomedicine, 2019, 63: 153009.》2.《Yuan Yang,Zheng Quan,Si Zhilin et al. Efficacy of Chinese Herbal Injections for Elderly Patients With pneumonia-A Bayesian Network Meta-analysis of Randomized Control Trials.[J] .Front Pharmacol, 2021, 12: 610745.》)

2. The ethical statement for meta-analysis should be added.

3. A series of statement are similar to existing findings. I suggest you add more content to your paper.

4. For full transparency and reproducibility, search strategies should include other strategies for the remaining database. Furthermore, the authors list the search strategy regarding text-term for Pubmed is too simple and incomprehensive, they mentioned they restricted their included studies in RCT while the search strategy was presented without study design.

5. Pubmed is not a database, but a search engine, please correct the association between pubmed and Medline.

6. The English writing needs to be improved. This manuscript is so hard to read due to some description is too simple.

7. Some statement is too absolute that should be interpreted cautiously, such as lines39 and lines197.

7. PLOS authors have the option to publish the peer review history of their article (what does this mean?). If published, this will include your full peer review and any attached files.

Reviewer #1: No

Reviewer #2: **Yes: **Jinghong Liang

---

## [Author Response · Author response to Decision Letter 0]

22 Oct 2021

We would like to express our sincere thanks to the reviewers for the constructive and positive comments.

Replies to Reviewer #1

1. The author restricted the search date up until April 2021, which’s been more than half year until now. Consider update this.

Answer: The search data has been updated to October, 2021 (line 24 and line 135) in the revised version.

2. The authors listed six CHIs and stated that “the head-to-head clinical trials comparing the efficacy of the six CHIs are lacking up to now.” If these six CHIs are the main ones of current use, it’s better to integrate these terms into the Search Strategy (i.e. specific CHIs as free text words)

Answer: We have added these terms into the Search Strategy and the whole strategy in PubMed are shown in Table 1. In addition, we have uploaded other strategies in remaining databases as a separate supporting information file.

3. Cochrane risk of bias tool is tailored for risk of bias assessment. Suggest use risk of bias throughout the manuscript instead of methodological quality, as these two could be different in some aspects.

Answer: We have use risk of bias instead of methodological quality in the revised version.

4. In the eligibility criteria, the primary outcome “clinical effective rate” is unclear. Would be good to see the exact definition. By the way, if SP is a leading infectious cause of death as the author said, why not to collect the mortality? Also, why not to collect the adverse event? How did the authors decide the importance of the outcomes for patients?

Answer: The exact definition of “clinical effective rate” has been added in Outcomes. We did only collect adverse drug reactions and adverse drug events will be included. The relative statements have been added in the revised version. As for outcomes, we decided the importance of the outcomes for patients by considering the outcomes of original studies, characteristics of the disease, clinical experiences and therapeutic advantage of the intervention. Actually, we considered 28 days mortality as the primary outcome at first. However, CHIs had more advantages in improving symptoms and signs as an adjuvant therapy. In addition, we found that there were few clinical studies reporting mortality by searching databases. Almost all studies reported the clinical effective rate, and we thought this outcome would be an alternative indicator to lower the death rate comparing with the mortality. As a result, we decided to collect clinical effective rate as the primary outcome. 

5. Not sure how will the authors use GRADE. Under “Evaluation of study quality and risk of bias”, it would be good to see a detailed description of the overall process. Also, NMA has more considerations than for conventional MA. From what the authors referred in citation 31, there is nothing about GRADE for NMA (the first paper was published in 2014 https://www.bmj.com/content/349/bmj.g5630). So it’s really unclear how will this work.

Answer: The statement of GRADE approach for NMA has been added in “Evaluation of study quality and risk of bias” and the correct reference has been cited in 31.

6. Under Data synthesis, “If data remains unobtainable, the reviewer will exclude the study”. This may not appropriate since this is not in your eligible criteria and will of course lead to more reporting bias. You may consider include and report them descriptively and explain why they are not included in the data synthesis.

Answer: The correct statement has been made in the Data synthesis.

7. Under “Measures of treatment effect.”: “All outcomes will be presented with their 95% credible intervals (95% CIs) and as well.” Is there anything missing?

Answer: The correction has been made in the revised version.

8. Under “Network meta-analysis” “After that, Publication bias will be reflected by a funnel plot” better to explain when is it appropriate to conduct a funnel plot, as the number of studies is not always enough for this analysis.

Answer: Correction has been made under “Assessment of publication bias” in the revised version.

9. For the limitations, why include only RCT is a limitation when qualified RCTs are among the highest level of evidence. Include Chinese and English may increase the “risk of bias”, do you mean “publication bias”?

Answer: Yes, the “risk of bias” means “publication bias” and correction has been made in the revised version. It’s our mistake to say only RCT is a limitation and we have deleted this sentence from Strengths and limitations of this study. 

10. The funders had and will not have a role in study design… Do you mean the funder had a role in the study design? Or it’s a mistake?

Answer: It seems that the paper requires to write that and I just copy it. When I submitted my manuscript, the paper required to include this sentence at the end of funding statement if the funders didn’t have a role in study design.

11. In the background, “The largest challenge in the next few years will be to diminish this high and unacceptable mortality rate.” Suggest delete as it didn’t add more useful information.

Answer: This sentence has been deleted in the revised version.

12. In the background, the authors mentioned the effectiveness and benefits of CHIs, what about the potential AEs of CHIs? Suggest to explain.

Answer: The statements of AEs of CHIs has been added in introduction in the revised version.

13. In the discussion, “The application of Chinese herbal injections could be promoted because of the risk of antibiotic resistance” is unclear. Do you mean because of the risk of antibiotic resistance from antibiotics? Otherwise, it reads like CHI has the risk.

Answer: Yes, it means because of the risk of antibiotic resistance from antibiotics. This sentence has been revised in the discussion.

14. What does “experimentally based evidence” mean in the Discussion?

Answer: It means “to generate reliable evidence based on a larger scale” and it has been revised in the revised version.

15. The authors stated “…this study will be the first Bayesian NMA of CHIs for the treatment of SP”, is there any existing frequentist NMA? Better to explain this point. If yes, may need to state why it’s necessary to do a new one.

Answer: There is not any existing frequentist NMA of CHIs for the treatment of SP by searching electronic databases. Maybe the statement of the first Bayesian NMA is too absolute and the meaning is unclear. So it has been corrected in the revised version.

16. Under the assessment of homogeneity, “I2≤ P≥0.1, 50%), If with homogeneity ( a fixed effect model will be adopted; If with obvious heterogeneity (P<0.01, I2 >50%), a random effect model will be applied.” is inappropriate. The choice between a fixed-effect and a random-effects meta-analysis should never be made on the basis of a statistical test for heterogeneity. Suggest refer to the Cochrane handbook. https://handbook-5-1.cochrane.org/chapter_9/9_5_4_incorporating_heterogeneity_into_random_effects_models.htm

Answer: The heterogeneity consists of clinical heterogeneity, methodological heterogeneity and statistical heterogeneity. The choice between a fixed-effect and a random-effects meta-analysis would not only be made on the basis of a statistical test for heterogeneity. Corrections have been made in the revised version.

17. Under “Assessment of similarity and consistency.”. What’s the method used to generate inconsistency factor? I didn’t find relevant information from citation 33. In addition, there is no consideration of the incoherence of each NMA and corresponding option to deal with the possible incoherence.

Answer: The loop-specific approach or node-splitting approach will be utilized to examine the loop inconsistency and the correct citations have been added in citation 33 and 34. The statement of incoherence has been added in the revised version.

18. There are several language problems, including grammar issue and the accuracy of the wording. For examples, 1) the conclusion in the abstract, “Our study findings will reveal which CHIs is the best” where CHIs should be CHI as the author assume there is only one best (By the way, how could we know there will not be two equal effective CHIs?); 2) “…can integrate direct evidence with indirect evidence from currently applied Chinese herbal injections into severe pneumonia (SP) to generate a clinically useful ranking …”, should from be “of”? What means “injections into SP (a disease)”? 3) mortality has already covered the meaning of “rate” so mortality itself is appropriate instead of “mortality rate” 4) mixed tense in one sentence such as “A fifth of the patients hospitalized for pneumonia need to be admitted to intensive care units (ICU), and a third of those require mechanical ventilation, while 21% of the patients from the community needed admission to the ICU and 26% of them needed mechanical ventilation. 5) “…the delay in the administration of adequate antimicrobials is clearly associated with mortality in patients…” do you mean “with higher mortality…”? 6) “criterias” under “Study selection and data extraction”. There are many others. This manuscript would benefit from further language editing.

Answer: Corrections have been made in the revised version and the English writing has been improved carefully.

Replies to Reviewer #2

1. This study lacks innovation. This manuscript is highly similarly to other systematic review, including disease types, analysis methods and research objects. A problem not fully addressed in discussion is mainly the similarity and the consistency evidence compared to the previous publications. I recommend that you can cite two studies for your information(1.《Huang Xingyue,Duan Xiaojiao,Zhu Yingli et al. Comparative efficacy of Chinese herbal injections for the treatment of community-acquired pneumonia: A Bayesian network meta-analysis of randomized controlled trials.[J] .Phytomedicine, 2019, 63: 153009.》2.《Yuan Yang,Zheng Quan,Si Zhilin et al. Efficacy of Chinese Herbal Injections for Elderly Patients With pneumonia-A Bayesian Network Meta-analysis of Randomized Control Trials.[J] .Front Pharmacol, 2021, 12: 610745.》)

Answer: Two systematic reviews have been conducted to assess the effectiveness of Chinese herbal injections for pneumonia. But participants included by one of the studies were mainly suffered from community-acquired pneumonia (CAP) whom were not needed to be hospitalized in ICU. As the worldwide spread of COVID-19, SP is given to more attention as its high mortality. Compared with CAP, SP is more serious and requires to be treated in ICU. Another study only included elderly patients with pneumonia. SP is associated with substantial morbidity and mortality, particularly in older adults or those with comorbid conditions. Nevertheless, SP can also occur in previously healthy young subjects originating distressing situations for its poor outcome. As for different disease severity and different ages of included participants, the CHIs being analyzed were different compared with the other two reviews. Our study will include three extra injections: Xuebijing injection, Shenfu injection and Shenmai injection which were widely used in either CAP or SP and proved to be effective by lots of high quality RCTs. In addition, for SP, patients were mostly in the stage of immunosuppression. According to the theory of traditional Chinese medicine, those patients were diagnosed with deficiency syndrome. Therefore, Shenfu injection and Shenmai injection which were not included by the two previous reviews will be included for analysis in our study. What’s more, the risk of bias will be evaluated with the Revised Cochrane risk-of-bias tool for randomized trials (ROB 2) and the overall quality of evidence for each outcome collected will be assessed using the GRADE which were not assessed in previous reviews. The statement has been added in discussion.

2. The ethical statement for meta-analysis should be added.

Answer: The ethical statement had been written in Ethics and dissemination under Methods.

3. A series of statement are similar to existing findings. I suggest you add more content to your paper.

Answer: We have added more contents like ADRs/ADEs of CHIs in introduction, the definition of clinical effective rate in outcomes, the GRADE approach in NMA, the distinction between our study and previous studies in the revised version.

4. For full transparency and reproducibility, search strategies should include other strategies for the remaining database. Furthermore, the authors list the search strategy regarding text-term for Pubmed is too simple and incomprehensive, they mentioned they restricted their included studies in RCT while the search strategy was presented without study design.

Answer: The search strategy of PubMed has been added completely in Table 1 and other strategies for the remaining database has been uploaded as a supporting information file.

5. Pubmed is not a database, but a search engine, please correct the association between pubmed and Medline.

Answer: We are very grateful for your teaching and have been clear about the association between PubMed and Medline.

6. The English writing needs to be improved. This manuscript is so hard to read due to some description is too simple.

Answer: The English writing has been improved carefully.

7. Some statement is too absolute that should be interpreted cautiously, such as lines39 and lines197.

Answer: Corrections have been made in the revised version.

---

## [Decision Letter · Decision Letter 1]

17 Dec 2021

PONE-D-21-21402R1Comparative efficacy of Chinese herbal injections for treating severe pneumonia: A protocol for systematic review and Bayesian network meta-analysis of randomized controlled trialsPLOS ONE

Dear Dr. Liu,

Thank you for submitting your manuscript to PLOS ONE. After careful consideration, we feel that it has merit but does not fully meet PLOS ONE’s publication criteria as it currently stands. Therefore, we invite you to submit a revised version of the manuscript that addresses the points raised during the review process.

ACADEMIC EDITOR: Please revise the manuscript and consider reviewer 1 comments

We look forward to receiving your revised manuscript.

Kind regards,

Ivan D. Florez, MD, MSc, PhD

Academic Editor

PLOS ONE

Journal Requirements:

Reviewers' comments:

Reviewer's Responses to Questions

**Comments to the Author**

1. Does the manuscript provide a valid rationale for the proposed study, with clearly identified and justified research questions?

Reviewer #1: Partly

Reviewer #2: Partly

2. Is the protocol technically sound and planned in a manner that will lead to a meaningful outcome and allow testing the stated hypotheses?

Reviewer #1: Partly

Reviewer #2: Partly

3. Is the methodology feasible and described in sufficient detail to allow the work to be replicable?

Reviewer #1: Yes

Reviewer #2: Yes

4. Have the authors described where all data underlying the findings will be made available when the study is complete?

Reviewer #1: Yes

Reviewer #2: Yes

5. Is the manuscript presented in an intelligible fashion and written in standard English?

Reviewer #1: Yes

Reviewer #2: Yes

6. Review Comments to the Author

You may also provide optional suggestions and comments to authors that they might find helpful in planning their study.

Reviewer #1: Thanks for authors response. There are still several issues need to be solved.

Still have some language issues. For example, lines 227-228 “…(CAP) whom were not all needed to be…”

Line 48 Authors focused on CHIs for severe pneumonia. CHIs are mostly used in China, I would suggest add background information about prevalence of pneumonia/ severe pneumonia in China or at least globally rather than just data in US.

Line 113 for the primary outcomes. I knew why authors choose clinical effective rate rather than mortality. But I would still say collecting data for mortality is important if this is an important outcome. If no study reported this, this should be clarified and discussed.

Line 134 under “Study selection and data extraction”:…screen the studies according to the inclusion criteria. The “inclusion criteria” is not accurate as you will refer to both inclusion and exclusion criteria. Consider “eligible criteria”. Same in line 136

Line 146 “Evaluation of study quality and risk of bias” What is study quality mean? If you use aOB to assess risk of bias then it’s risk of bias. There are still mixing use of study quality and risk of bias in the method. If you used GRADE to assess certainty of evidence (more used in latest GRADE series than “quality of evidence”), then it’s certainty of the evidence. Please refer to https://www.bmj.com/content/371/bmj.m3900.abstract. Also, authors may consider how to draw the conclusion according to the GRADE approach (article in same link).

Line 175, authors considered statistically significant. Will there any consideration on clinical significant based on minimal important difference (MID)? Better to clarify.

Lines 221-223 Need citations to support the sentence “a large number of studies…”

Lines 224-225 The results are yet to come, how could we know CHIs “could be widely used because of its efficacy and safety”? Even when you have data that shows very good benefit in your final NMA, you will still need to consider the certainty of the evidence to give the conclusion cautiously.

Lines 226-228: according to authors’ description, SP is covered by CAP, which could also be seen in the references 28-29. It seems to me that the NMA in 2019 by Huang X et al has already included all the CAP populations and you are selecting a group of patients from the whole bunch. I did not see a high value of this point now.

Lines 236-237 “…proved to be effective by lots of high quality RCTs[15-17, 44]”. What does this “high quality” mean? Have you assessed them already? I would not say any quality thing without any assessment.

Reviewer #2: Thank you for addressing most of the reviewers' comments and improving English. My comments were addressed by the authors and I don't have further suggestions.

7. PLOS authors have the option to publish the peer review history of their article (what does this mean?). If published, this will include your full peer review and any attached files.

Reviewer #1: **Yes: **Xiaoqin Wang

Reviewer #2: **Yes: **Jinghong Liang

---

## [Author Response · Author response to Decision Letter 1]

20 Dec 2021

We would like to express our sincere thanks to the reviewers for the constructive and positive comments.

Replies to Reviewer 1

1. lines 227-228 “…(CAP) whom were not all needed to be…”

Answer: The correction has been made in the revised version and the English writing has been improved carefully.

2. Line 48 Authors focused on CHIs for severe pneumonia. CHIs are mostly used in China, I would suggest add background information about prevalence of pneumonia/ severe pneumonia in China or at least globally rather than just data in US.

Answer: The prevalence of pneumonia/severe pneumonia in China is deficiency. The only data was about the prevalence of pneumonia in 2 weeks recorded in China Health Statistics Yearbook in 2013 and we have added this content in introduction. 

3. Line 113 for the primary outcomes. I knew why authors choose clinical effective rate rather than mortality. But I would still say collecting data for mortality is important if this is an important outcome. If no study reported this, this should be clarified and discussed.

Answer: The 28-day mortality has been added in the revised version as one of the primary outcomes. If we find that there are not enough original studies reporting 28-day mortality when we conduct this NMA, we will discuss and clarify this problem.

4. Line 134 under “Study selection and data extraction”:…screen the studies according to the inclusion criteria. The “inclusion criteria” is not accurate as you will refer to both inclusion and exclusion criteria. Consider “eligible criteria”. Same in line 136

Answer: The correction has been made in the revised version.

5. Line 146 “Evaluation of study quality and risk of bias” What is study quality mean? If you use aOB to assess risk of bias then it’s risk of bias. There are still mixing use of study quality and risk of bias in the method. If you used GRADE to assess certainty of evidence (more used in latest GRADE series than “quality of evidence”), then it’s certainty of the evidence. Please refer to https://www.bmj.com/content/371/bmj.m3900.abstract. Also, authors may consider how to draw the conclusion according to the GRADE approach (article in same link).

Answer: The correction has been made in the revised version and the details about how to draw the conclusion according to the GRADE approach have been added in the revised version.

6. Line 175, authors considered statistically significant. Will there any consideration on clinical significant based on minimal important difference (MID)? Better to clarify.

Answer: Minimal important difference (MID) is mainly used to explain the results of scores or scales. However, our primary outcomes are 28-day mortality and clinical effective rate, which do not involve scores or scales. Moreover, we don’t find recognized MID of our outcomes by searching databases. Hence, we don’t consider the clinical significant based on minimal important difference.

7. Lines 221-223 Need citations to support the sentence “a large number of studies…”

Answer: The relevant citations have been added in the revised version.

8. Lines 224-225 The results are yet to come, how could we know CHIs “could be widely used because of its efficacy and safety”? Even when you have data that shows very good benefit in your final NMA, you will still need to consider the certainty of the evidence to give the conclusion cautiously.

Answer: This sentence has been deleted in discussion.

9. Lines 226-228: according to authors’ description, SP is covered by CAP, which could also be seen in the references 28-29. It seems to me that the NMA in 2019 by Huang X et al has already included all the CAP populations and you are selecting a group of patients from the whole bunch. I did not see a high value of this point now.

Answer: Actually, we aim to express that SP is more serious than CAP and therefore the CHIs utilized between them are different. Maybe it is repetitive with the following contents. Hence, we decide to delete these sentences after considering your advice.

10. Lines 236-237 “…proved to be effective by lots of high quality RCTs[15-17, 44]”. What does this “high quality” mean? Have you assessed them already? I would not say any quality thing without any assessment.

Answer: We have deleted “high quality” in this sentence.

---

## [Decision Letter · Decision Letter 2]

5 Jan 2022

Comparative efficacy of Chinese herbal injections for treating severe pneumonia: A protocol for systematic review and Bayesian network meta-analysis of randomized controlled trials

PONE-D-21-21402R2

Dear Dr. Liu,

We’re pleased to inform you that your manuscript has been judged scientifically suitable for publication and will be formally accepted for publication once it meets all outstanding technical requirements.

Kind regards,

Ivan D. Florez, MD, MSc, PhD

Section Editor

PLOS ONE

Additional Editor Comments (optional):

Reviewers' comments:

Reviewer's Responses to Questions

**Comments to the Author**

1. Does the manuscript provide a valid rationale for the proposed study, with clearly identified and justified research questions?

Reviewer #1: Yes

2. Is the protocol technically sound and planned in a manner that will lead to a meaningful outcome and allow testing the stated hypotheses?

Reviewer #1: Yes

3. Is the methodology feasible and described in sufficient detail to allow the work to be replicable?

Reviewer #1: Yes

4. Have the authors described where all data underlying the findings will be made available when the study is complete?

Reviewer #1: Yes

5. Is the manuscript presented in an intelligible fashion and written in standard English?

Reviewer #1: Yes

6. Review Comments to the Author

You may also provide optional suggestions and comments to authors that they might find helpful in planning their study.

Reviewer #1: I have no additional comments. Just one suggestion to authors' response letter: make the response self-contained (rule 4 here https://journals.plos.org/ploscompbiol/article?id=10.1371/journal.pcbi.1005730)

7. PLOS authors have the option to publish the peer review history of their article (what does this mean?). If published, this will include your full peer review and any attached files.

Reviewer #1: **Yes: **Xiaoqin Wang

---

## [Editor Report · Acceptance letter]

13 May 2022

PONE-D-21-21402R2 

Comparative efficacy of Chinese herbal injections for treating severe pneumonia: A protocol for systematic review and Bayesian network meta-analysis of randomized controlled trials 

Dear Dr. Liu:

I'm pleased to inform you that your manuscript has been deemed suitable for publication in PLOS ONE. Congratulations! Your manuscript is now with our production department. 

Kind regards, 

on behalf of

Dr. Ivan D. Florez 

Section Editor

PLOS ONE